# Longevity of different in-office treatments for dentin hypersensitivity: A 6-month randomized and parallel clinical trial

Fernanda de Souza e Silva Ramos[1], André Luiz Fraga Briso[1‡], Érika Mayumi Omoto[1], Edgar Dutra Zanotto[2‡], Paulo Henrique dos Santos[3‡], Bruna Perazza[1], Ticiane Cestari Fagundes[1‡]*

1 Department of Restorative Dentistry, São Paulo State University (UNESP), Araçatuba, São Paulo, Brazil, 2 Department of Materials Engineering, Federal University of São Carlos (UFSCar), São Carlos, São Paulo, Brazil, 3 Faculty of Dentistry, University of Toronto, Toronto, Ontario, Canada

☉ These authors contributed equally to this work.
‡ These authors also contributed equally to this work.
* ticiane.fagundes@unesp.br

## Abstract

This longitudinal, randomized, parallel-design clinical trial aimed to evaluate the long-term effectiveness of different in-office treatments for dentin hypersensitivity (DH) over a 6-month period. A total of 192 teeth presenting DH associated with root surface exposure were treated with one of four desensitizing agents: fluoride varnish (Duraphat – FLU, active control), bioactive ceramic solution (Biosilicate – BIOS), universal self-etching adhesive (Single Bond Universal – SBU), and a bioactive, photoactivated varnish containing PRG fillers (SPRG). DH was assessed using a visual analogue scale (VAS) to analyse intensity of sensitivity and a computerized visual scale (CoVAS) to observe the time course of sensibility at baseline, 7, 15, 30 days and 6 months post-treatment. Comparisons between groups were conducted using Fisher's exact test for qualitative variables and analysis of variance (ANOVA) for quantitative variables. Longitudinal outcomes (VAS and CoVAS) were evaluated using linear mixed-effects models, with post hoc comparisons performed using orthogonal contrasts. Statistical differences were found for VAS analysis (p < 0.05). Comparing the desensitizers, FLU showed higher sensitivity than SBU, after 7 days. Comparing follow-up periods, a significant reduction in DH beginning at 7 days for BIOS and SBU, and at 15 days for FLU and SPRG. Regarding CoVAS time course, FLU exhibited a significantly shorter time course than SBU, after 15 days and 6 months of follow-up (p ≤ 0.05). SPRG was the only treatment that showed no difference in reaction time between 6 months and baseline (p > 0.05). In conclusion, root exposure height and the presence of an antagonist tooth may influence the initial intensity of DH. All treatments reduced DH over 6 months. Only bioactive resin varnish maintained the same reaction time in all periods. Universal adhesive system had slowed pain reaction time compared to conventional fluoride varnish.

**Data availability statement:** All relevant data are within the manuscript and its Supporting Information files.

**Funding:** This study was financed in part by the Coordenação de Aperfeiçoamento de Pessoal de Nível Superior – Brasil (CAPES) -Finance Code 001. This research was also supported by the São Paulo Research Foundation—FAPESP [research grant number 2020/07625-0 and scholarship grant number 2020/07443-9]. The funders had no role in study design, data collection and analysis, decision to publish, or preparation of the manuscript.

**Competing interests:** The authors have declared that no competing interests exist.

## Introduction

Dentin hypersensitivity (DH) is a frequent clinical condition and a notable concern in public oral health, as it negatively affects patients' comfort and quality of life [1]. Clinically, DH is characterized by a sudden, short-lasting pain triggered by thermal, tactile, chemical, or osmotic stimuli [1–6]. It occurs when dentin is exposed, and the dentinal tubules remain open to the oral environment [1–3]. The most accepted explanation for this pain mechanism is the hydrodynamic theory, which proposes that external stimuli cause shifts in the flow of fluid inside the tubules, leading to the activation of nerve fibers around odontoblasts, resulting in pain [3,4,7].

Eliminating etiological causes is the first step in treating DH [2]. In this context, treatments that promote occlusion of the dentinal tubules offer a more direct approach by decreasing dentin permeability and have shown promising results in clinical practice [3]. Among the available in-office therapies, fluoride varnishes are widely used due to their ability to form calcium fluoride-like deposits that block the entrance to dentinal tubules, thereby reducing sensitivity [8–10]. However, their action tends to be short-lived, often requiring repeated applications to maintain desensitizing effects [8]. An alternative approach involves the use of bioactive materials, such as bioactive glasses, which exhibit regenerative potential and have the capacity to induce the formation of mineralized barriers [3,7,11–14]. In particular, Biosilicate has been associated with dentinal tubule occlusion, stimulation of remineralization, and demineralization prevention [7,11–14].

Photoactivated materials also offer effective options for managing DH. Universal adhesives, for example, can seal dentin by forming a hybrid layer, which stops fluid movement and immediately reduces the symptoms. [15–17] Additionally, a light-cured resin-based varnish incorporating surface pre-reacted glass-ionomer filler technology (S-PRG) was developed [18,19]. This multifunctional bioactive material, incorporated into a polyacid resin matrix, gradually releases ions such as fluoride, strontium, borate, aluminum, silicate, and sodium that contribute to tubule occlusion and DH relief [19–21]. Recent clinical evidence–based scoping review concluded that materials containing S-PRG fillers have shown generally favorable clinical outcomes across various desensitizing treatments. [21].

Despite the range of products available, DH remains one of the most reported causes of dental pain and is often challenging to manage over time [22]. Recent systematic reviews emphasize that no single therapeutic approach can currently be considered the gold standard [1,2,8,23]. Although a study supports the short-term effectiveness of in-office desensitizing agents [4], there is still a lack of high-quality randomized clinical trials investigating their clinical stability over extended periods of analysis [4,24]. Furthermore, follow-up clinical trials extending beyond six months are recommended to accurately determine the longevity of therapeutic effects [8].

Thus, the present randomized and parallel clinical trial aimed to evaluate the durability of different in-office desensitizing treatments after a 6-month follow-up. Additionally, the influence of characteristics related to root exposure on the initial intensity of dentin hypersensitivity was also evaluated. The null hypotheses tested were that 1) the characteristics of patients and teeth, harmful habits, previous oral treatments,

and hygiene instructions would not influence the intensity of initial sensibility; 2) there is no difference in the reduction of DH over time, for each in-office treatment; 3) there is no difference in the efficacy of different in-office treatments for DH, regardless of the experimental times analyzed.

## Materials and methods

### Experimental design and ethical aspects

This clinical trial was approved by the local ethics committee in human research (#30122220.1.0000.5420; S1-S6 Files) and registered in clinical trial database (RBR-5nw4zp; S7 File). The study was designed according to the Consolidated Standards of Reporting Trials (CONSORT; S8 File). The research was planned as interventional, randomized, prospective, single-center, single-blind (participants), and parallel clinical trial. Further details of these procedures can be found in our previous paper [25].

### Sample size, subjects' selection and groups

Sample size calculation was based on data from a previous study [26], using analysis of variance (ANOVA) sample size (Sigma Plot 12.0, Systat Software Inc., London, UK). The analysis assumed a minimum detectable difference in means of 0.210 (expected standard deviation of residuals of 0.3) on the Visual Analogue Scale (VAS), with a significance level (α) of 0.05, desired power of 0.73, and estimated dropout rate of 20.0%. The calculation resulted in initial sample size of 48 teeth per group. Participants were recruited from the undergraduate clinic at the local Faculty of Dentistry.

Inclusion criteria were adults aged 20–60, in good general health, with no allergies to dental products, and at least one non-cavitated or minimally cavitated root exposure (≤1 mm) with hypersensitivity ≥ grade 3 (VAS). For this analysis, relative isolation with cotton rolls was performed. DH was assessed using an air jet stimulus applied with a triple syringe positioned 1 cm from the cervical area of the tooth for 5 seconds. Immediately afterward, patients rated their pain on a 10-cm horizontal VAS line, marking a vertical line to indicate perceived sensitivity. The scale was interpreted as follows: 0 = no pain; 1–3 = mild pain; 4–6 = moderate pain; and 7–10 = severe pain [27,28].

Exclusion criteria were pregnancy or nursing, smoking, active caries or periodontal disease, recent use of desensitizing agents, orthodontic or removable partial denture use on the target tooth, and use of analgesics or anti-inflammatories.

A total of 68 patients were screened; 42 patients (n = 192 teeth) were included. Participants were randomly assigned to one of four treatment groups: fluoride varnish (FLU, active control), biosilicate solution (BIOS), universal adhesive system (SBU), or varnish containing S-PRG particles (SPRG), (Table 1) [25].

### Clinical exams, randomization, and baseline evaluations

Informed consents were obtained from all participants. Patients completed a questionnaire to identify personal characteristics and potentially harmful habits, including lifestyle factors, dietary practices, previous oral treatments and oral hygiene behaviors that could be associated with DH [29]. The oral health status of each patient was assessed using the decayed, missing, and filled teeth index (DMFT), visible plaque index (VPI), and gingival bleeding index (GBI).

Subsequently, clinical measurements of depth and height of root exposure were obtained using a millimeter probe. The height of root exposure was measured as the distance from the most apical point of the cementoenamel junction to the highest point of the free marginal gingiva; no upper limit was defined for this measurement [30]. The height of root exposure and the VAS sensitivity score were treated as stratification variables during the randomization process. The tooth was considered the unit of analysis.

The randomized allocation sequence was generated by the operator of this study, using the Microsoft Excel software (Microsoft Corp., Redmond, WA, USA) with the stratified randomization method. Eligible teeth were recorded with respect to sensitivity and exposure height. Teeth were categorized according to their VAS score and exposure height into four

**Table 1. Commercial name, manufacturer, batch number, composition, and mode of application of in-office desensitizing materials used in this study.**

| Material | Manufacturer | Batch# | Composition | Mode of application |
|---|---|---|---|---|
| Duraphat (FLU) | Colgate-Palmolive Company, New York, NY, USA | 022001 | 5% NaF (22.600 ppm), colophony; solvent, shellac; mastic; saccharine and others. | A thin layer was passively applied under the clean and dry surface with a disposable applicator, remaining for 5 min. A second layer was applied, remaining for another 5 min.* |
| Biosilicate(BIOS) | Laboratory of Vitreous Materials, Federal University of São Carlos, São Carlos, SP, Brazil | – | The solution was composed of Biosilicate powder ($P_2$; $O_5$-$Na_2$; O-CaO-$SiO_2$ 1–10 μm) and distilled water (1:10 ratio) and for simulation of the professional-use products, the particles were mixed immediately before application | A thin layer was applied for 5 s on the clean and dry surface with a disposable applicator, remaining stable for 10 min. § |
| Single Bond Universal (SBU) | 3M ESPE, St. Paul, MN, USA | 1833100782 | BISGMA; HEMA; UDMA; DPIHFP, 10-MDP; solvent; water; silane; and others. | A two thin layer was actively applied for 20 s on the clean and dry surface with a disposable applicator and light-cured for 10 s. No previous acid etching was performed.* |
| Barrier Coat (S-PRG filler varnish –SPRG) | Shofu INC, Kyoto, Japan. | 121901 | S-PRG filler (3.0μm): TEGDMA; Bis-MPEPP; fluorine boron aluminosilicate; MAA; phosphonic acid; and others. | Two thin layers of one drop active mixed with base were applied for 3s on the clean and dry surface with specific applicator and light-cured for 10s. The uncured layer was removed from surface with a water-moistened cotton pellet.* |

Abbreviations: NaF (Sodium fluoride); $P_2$ (Posphorus); O (Oxygen); CaO (Calcium Oxide); $SiO_2$ (Silicon dioxide) TMPnano (nanoparticulate sodium trimetaphosphate) TEGDMA (triethylene glycol dimethacrylate); BISGMA (diglycidildimethacrylate A); HEMA (Hydroxyethylmethacrylate); UDMA (1,3 glycol dimethacrylate) DPIHFP (Diphenyliodonium hexafluorophosphate); 10-MDP (10-decanediol phosphate methacrylate); Bis-MPEPP (bisphenol A polyethoxy methacrylate); MAA (methacrylic acid). * According to the manufacturer. § de Castro Oliveira et al. [20].

strata: (a) low sensitivity (VAS 3–7) and short exposure height (≤3 mm); (b) low sensitivity (VAS 3–7) and greater exposure height (>3 mm); (c) high sensitivity (VAS 8–10) and short exposure height (≤3 mm); and (d) high sensitivity (VAS 8–10) and greater exposure height (>3 mm). Patients who had more than one eligible tooth, classification was based on the mean of exposure heights and the mean of VAS sensitivity scores.

Computerized visual scale (CoVAS, Medoc; Ramat Yishai, Northern District, Israel) measurements were performed by applying a constant air-jet stimulus to the buccal surface of the tooth at a 10-mm distance for up to 30 seconds. During stimulation, the participant continuously adjusted a manually controlled potentiometer to indicate the perceived sensibility on a 0–100 scale [31]. For each assessment, the maximum sensibility score and the exact time (in seconds) were recorded. Additionally, if the participant reached a score of 100, the stimulus was immediately discontinued and the corresponding time was recorded. For example, if a participant reported a maximum score of 13 at 27 seconds and this value remained constant until 30 seconds, the recorded values were 13 and 27 seconds. However, if a participant reached a pain score of 100 at 1 second the stimulus was stopped immediately, and these two values were registered.

## Blinding

The participants were blinded to the treatments assigned to the groups, because the application procedures for the desensitizing materials differed, the operator could not be blinded. The desensitizing procedures were carried out by an operator, without revealing the type of material to the participants.

## Interventions

After the previous evaluations, prophylaxis was performed, and a buccal retractor was positioned to separate the lips and cheek. The tooth that received the desensitizer was isolated with cotton rolls and dried with a jet of air, with the humidity

being controlled with a sucker. All protocol steps of each in-office desensitizer treatment were described in Table 1. The FLU desensitizer was applied in a once-weekly session, for three weeks. The BIOS solution was mixed using 0.15 mg of powder with 1.35 mL of distilled water and applied such as the previous desensitizer [13]. Light-curing products (SBU and SPRG) were applied and light-cured (1050 mW/cm$^2$, LED Radii-cal, SDI Brazil Industry and Commerce LTDA, SP, Brazil) only in the first section. Patients were instructed regarding oral hygiene instructions, being advised the non-use of tooth-paste and mouthwash with desensitizing composition. Furthermore, the patients received all the necessary complementary dental treatments.

## Outcomes

The intensity of DH as well as the sensibility time course was evaluated before treatments (baseline) and after 7, 15, 30 days and 6 months. VAS and CoVAS were used. DH analysis was measured before the reapplication for FLU and BIOS groups, in each session.

## Statistical analysis

Comparisons involving qualitative variables among groups were performed using Fisher's exact test, as well as comparisons involving quantitative variables were performed using ANOVA (Tables 2–4). The influence of those initial data in intensity of sensibility (VAS) was analyzed by linear regression model with post-test using orthogonal contrasts. Comparisons of quantitative longitudinal outcomes (VAS and CoVAS time course), linear mixed-effects regression models including fixed and random effects were employed (Table 5 and Fig 2). A variance components covariance structure was adopted. All models were adjusted for the initial stratification variables used during randomization, ensuring adequate

**Table 2. Descriptive results from characteristics of patients, described as mean (±standard deviation) or number (percentage %) per group, after 6 months.**

| Groups | FLU | BIOS | SBU | SPRG | p value | Test |
|---|---|---|---|---|---|---|
| Number of patients | 8 | 11 | 9 | 9 | – | |
| *Stratified randomization* | | | | | | |
| Strata (a) | 8 (100%) | 11 (100%) | 5 (55.6%) | 8 (88.9%) | 0.01 | Fisher |
| Strata (b) | 0 (0%) | 0 (0%) | 1 (11.1%) | 1 (11.1%) | | |
| Strata (c) | 0 (0%) | 0 (0%) | 3 (33.3%) | 0 (0%) | | |
| Strata (d) | 0 (0%) | 0 (0%) | 0 (0%) | 0 (0%) | | |
| *Demographic characteristics* | | | | | | |
| Age | 41.5 (±8.4) | 39.9 (±11.7) | 42.9 (±10.8) | 45.7 (±6.7) | 0.62 | ANOVA |
| Male | 1 (12.5%) | 6 (54.5%) | 2 (22.2%) | 4 (44.4%) | 0.23 | Fisher |
| Female | 7 (87.5%) | 5 (45.5%) | 7 (77.8%) | 5 (55.6%) | | |
| *Oral conditions* | | | | | | |
| DMFT | 10.5 (±5.7) | 13.3 (±4.8) | 10.8 (±7.7) | 13.1 (±6.2) | 0.66 | ANOVA |
| VPI | 8.2 (±3.4) | 5.3 (±2.8) | 5.2 (±3.3) | 5.5 (±2.0) | 0.11 | |
| GBI | 0.1 (±0.2) | 0 (0) | 0 (0) | 0 (0) | – | |
| *Pattern of disocclusion* | | | | | | |
| Canine guidance | 6 (75%) | 9 (81.8%) | 8 (88.9%) | 5 (55.6%) | 0.43 | Fisher |
| Group function | 2 (25%) | 2 (18.2%) | 1 (11.1%) | 4 (44.4%) | | |
| *Antagonist teeth* | | | | | | |
| No | 0 (0%) | 1 (9.1%) | 0 (0%) | 0 (0%) | 0.99 | Fisher |
| Yes | 8 (100%) | 10 (90.9%) | 9 (100%) | 9 (100%) | | |

*Abbreviations: FLU (fluoride varnish); BIOS (biosilicate solution); SBU (universal adhesive system); SPRG (varnish containing S-PRG particles).*

**Table 3. Descriptive results from characteristics of teeth, described as mean (±standard deviation) or number (percentage %) per group, after 6 months.**

| Groups | FLU | BIOS | SBU | SPRG | p value | Test |
|---|---|---|---|---|---|---|
| *Root exposure measurements* | | | | | | |
| Height | 1.9 (±1.5) | 2.1 (±1.1) | 1.5 (±1.1) | 2 (±1.0) | 0.53 | ANOVA |
| Depth | 0.3 (±0.2) | 0.3(±0.2) | 0.5 (±0.3) | 0.4 (±0.3) | 0.46 | |
| *Type of tooth* | | | | | | |
| Canine | 3 (37.5%) | 4 (36.4%) | 1 (11.1%) | 1 (11.1%) | 0.54 | Fisher |
| Central Incisor | 0 (0%) | 0 (0%) | 1 (11.1%) | 0 (0%) | | |
| Lateral incisor | 0 (0%) | 0 (0%) | 0 (0%) | 1 (11.1%) | | |
| Molar | 4 (50%) | 5 (45.5%) | 5 (55.6%) | 7 (77.8%) | | |
| Premolar | 1 (12.5%) | 2 (18.1%) | 2 (22.2%) | 0 (0%) | | |

*Abbreviations: FLU (fluoride varnish); BIOS (biosilicate solution); SBU (universal adhesive system); SPRG (varnish containing S-PRG particles).*

**Table 4. Descriptive results of harmful habits and oral treatments/instructions, described as number (percentage %) per group, after 6 months.**

| Groups | | FLU | BIOS | SBU | SPRG | p value | Test |
|---|---|---|---|---|---|---|---|
| *Harmful habits* | | | | | | | |
| Do you have gastric problem with reflux? | No | 6 (75%) | 7 (63.64%) | 8 (88.89%) | 8 (88.89%) | 0.48 | Fisher |
| | Yes | 2 (25%) | 4 (36.36%) | 1 (11.11%) | 1 (11.11%) | | |
| How often during the week do you have gastric problems with reflux? | Yes, once a week | 2 (25%) | 4 (36.36%) | 0 (0%) | 0 (0%) | 0.11 | |
| | Yes, every day | 0 (0%) | 0 (0%) | 1 (11.11%) | 1 (11.11%) | | |
| Do you consume acidic foods or drinks? | No | 1 (12.5%) | 1 (9.09%) | 2 (22.22%) | 0 (0%) | 0.60 | |
| | Yes | 7 (87.5%) | 10 (90.91%) | 7 (77.78%) | 9 (100%) | | |
| How often do you consume acidic foods or drinks during the week? | Yes, once a week | 1 (12.5%) | 0 (0%) | 0 (0%) | 1 (11.11%) | 0.78 | |
| | Yes, 2 × a week | 0 (0%) | 0 (0%) | 1 (11.11%) | 2 (22.22%) | | |
| | Yes, 3 × a week | 1 (12.5%) | 3 (27.27%) | 1 (11.11%) | 1 (11.11%) | | |
| | Yes, every day | 5 (62.5%) | 7 (63.64%) | 5 (55.56%) | 5 (55.56%) | | |
| Do you have dental clenching? | No | 6 (75%) | 4 (36.36%) | 5 (55.56%) | 5 (55.56%) | 0.46 | |
| | Yes | 2 (25%) | 7 (63.64%) | 4 (44.44%) | 4 (44.44%) | | |
| Do you have bruxism? | No | 5 (62.5%) | 7 (63.64%) | 7 (77.78%) | 6 (66.67%) | 0.93 | |
| | Yes | 3 (37.5%) | 4 (36.36%) | 2 (22.22%) | 3 (33.33%) | | |
| Do you usually put objects in your mouth? | No | 7 (87.5%) | 10 (90.91%) | 9 (100%) | 8 (88.89%) | 0.89 | |
| | Yes | 1 (12.5%) | 1 (9.09%) | 0 (0%) | 1 (11.11%) | | |
| *Previous oral treatments and instructions* | | | | | | | |
| Have you ever had periodontal treatment? | No | 1 (12.5%) | 0 (0%) | 0 (0%) | 2 (22.22%) | 0.19 | Fisher |
| | Yes | 7 (87.5%) | 11 (100%) | 9 (100%) | 7 (77.78%) | | |
| Have you ever received oral hygiene instructions? | No | 0 (0%) | 0 (0%) | 1 (11.11%) | 0 (0%) | 0.70 | |
| | Yes | 8 (100%) | 11 (100%) | 8 (88.89%) | 9 (100%) | | |

*Abbreviations: FLU (fluoride varnish); BIOS (biosilicate solution); SBU (universal adhesive system); SPRG varnish containing S-PRG particles).*

control of these factors in the study design. Model assumptions, particularly the normality of residuals, were evaluated through graphical inspection using histograms, quantile–quantile (Q–Q) plots, and residual-versus-fitted plots. Group and time comparisons were performed using post hoc orthogonal contrasts. No additional correction for multiple comparisons was applied. Statistical analyses were conducted using SAS 9.4 software (SAS Institute Inc. NC, USA).

**Table 5. Mean (± standard of deviation) presented by different desensitizing materials according to the VAS scale.**

| Material *Time* | FLU | BIOS | SBU | SPRG |
|---|---|---|---|---|
| *Baseline* | 6.0±2.0 Ad | 5.5±2.6 Ac | 6.3±2.3 Ab | 5.5±2.3 Ac |
| *7 days* | 5.7±1.7 Bcd | 4.0±2.9 ABb | 4.5±2.9 Aa | 4.9±2.5 ABbc |
| *15 days* | 4.9±2.1 Abc | 4.2±3.0 Ab | 4.1±2.7 Aa | 4.2±2.9 Aab |
| *30 days* | 4.4±2.5 Aab | 3.4±3.3 Aab | 3.7±2.9 Aa | 3.9±2.5 Aab |
| *6 months* | 3.6±2.5 Aa | 2.9±2.6 Aa | 4.6±3.7 Aa | 3.8±2.6 Aa |

*Abbreviations: FLU (fluoride varnish); BIOS (biosilicate solution); SBU (universal adhesive system); SPRG (varnish containing S-PRG particles). Uppercase letters indicate significant differences among materials within the same analysis times. Lowercase letters indicate significant differences among analysis times within the same material. Linear mixed-effects regression models including fixed and random effects were employed (p ≤ 0.05).*

A per-protocol analysis was conducted, including only participants who completed all evaluation timepoints. All data were considered statistically different if $p ≤ 0.05$).

## Results

This clinical trial was conducted from July 2021 to March 2023. Participant recruitment occurred between May 28, 2021, and February 17, 2022 (last follow-up March 2023). The flowchart showing the distribution of patients is presented in Fig 1. The distribution per group of descriptive results about characteristics of patients and teeth are presented in Table 2 and 3, respectively. The distribution per group of answers about harmful habits, previous oral treatments, and hygiene instructions are presented in Table 4. No significant difference was found when distributions among groups of above data were analyzed, after 6 months ($p > 0.05$). The regression model found that some descriptive data influenced DH: depth of root exposure only for VAS, and presence of antagonist teeth for both scales ($p ≤ 0.05$).

The presence of antagonist teeth promoted more DH because 53% of the teeth with antagonists presented VAS above 6, while in teeth without antagonists, these values were observed in 28% of the teeth, respectively.

In Table 5, FLU showed higher sensitivity than SBU, after 7 days ($p < 0.05$), while other groups showed no statistical differences ($p > 0.05$) for VAS data. Statistically differences were found in the reduction of DH over time when each in-office treatment was evaluated for both sensibility analysis ($p ≤ 0.05$). According to VAS data, FLU and BIOS showed a reduction in sensitivity at 15 days and 7 days, respectively; and both desensitizers promoted DH stability at 30 days until 6 months. SBU demonstrated a significant decrease after 7 days, which persisted through 6 months. SPRG presented a reduction at 15 days, remaining until 6 months.

Fig 2 illustrates that the desensitizing treatments generated response curves with identifiable peaks, representing the CoVAS time course corresponding to the highest intensity of sensitivity reported by the patient, measured in seconds. FLU exhibited a significantly shorter time course than SBU, after 15 days and 6 months of follow-up ($p ≤ 0.05$). Comparing the follow-up periods, FLU showed a shorter CoVAS time course for all periods when compared to baseline ($p ≤ 0.05$). Baseline CoVAS time course was shorter than 15 days and 6 months, for BIOS ($p ≤ 0.05$); same results were found comparing 7 and 30 days with 6 months. SBU showed longer CoVAS time course at the 6-month follow-up, when compared to baseline and 30 days ($p ≤ 0.05$). In contrast, SPRG did not present statistically significant differences among the follow-up periods.

## Discussion

The impact of DH on patients' daily routine has led researchers to search for an effective and long-lasting treatment [4,5]. Before evaluating the effectiveness of desensitizing treatments, it is also important to highlight that DH is a subjective

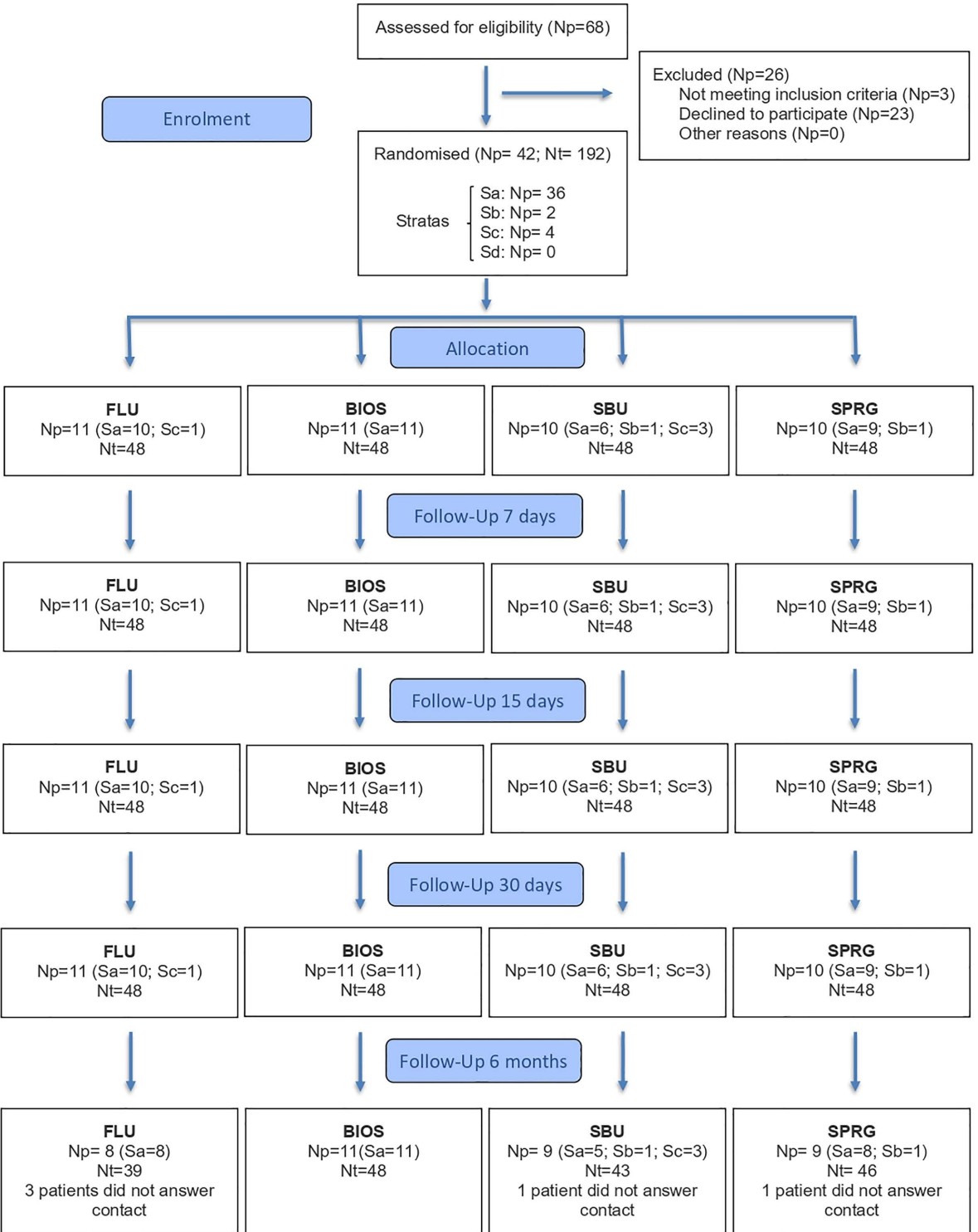

**Fig 1. Consort flowchart.** FLU: fluoride varnish, BIOS: biosilicate solution, SBU: universal adhesive system, SPRG: varnish containing S-PRG particles, Np: number of patient, Nt: number of teeth.

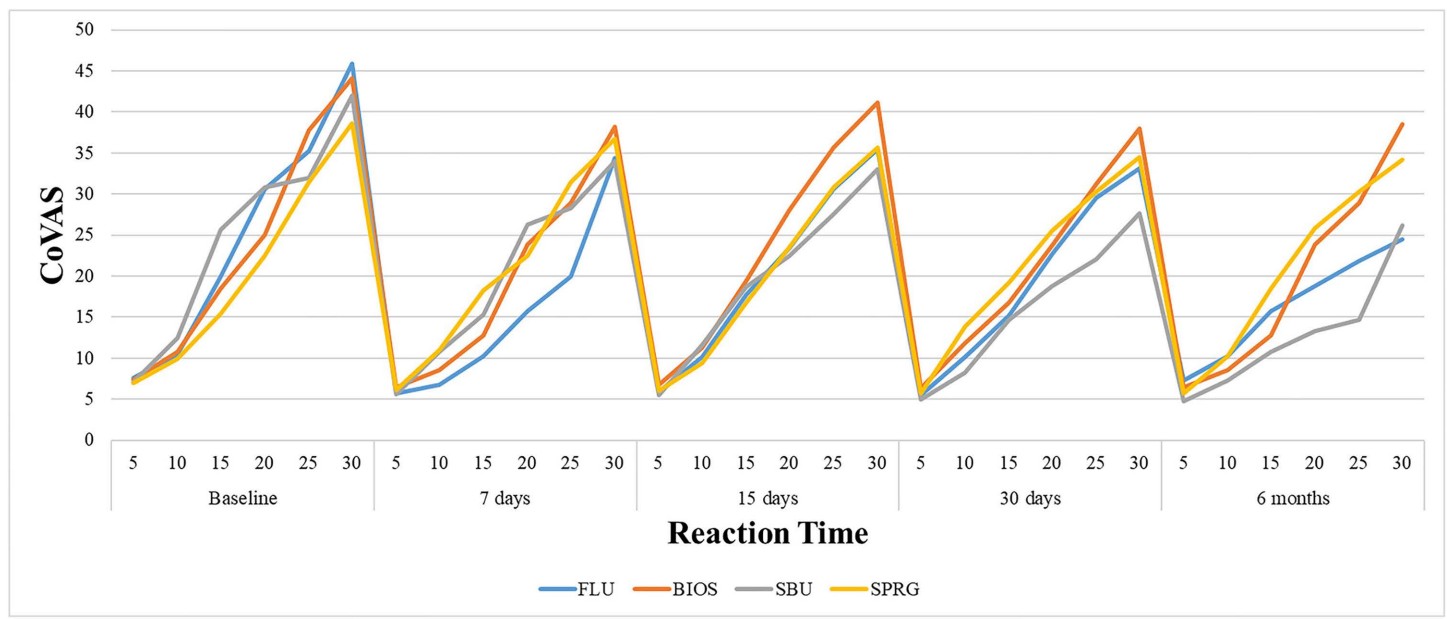

**Fig 2. Curves with peaks showing the CoVAS time course of sensitivity by group and time of evaluation.**

condition that is difficult to quantify [4]. VAS and CoVAS analysis were chosen because of their easy application and good patient tolerance [28]. VAS was also selected due to its wide use in literature [4,13,16–18,24,27,28,31–33]; further-more, CoVAS was added because it is computerized, allowing monitoring of patient's sensibility time course [28,31]. The evaporative method was used as it is more precise than the tactile method for assessing DH, in addition to promoting the stimulation of a larger area of dentin [34,35].

Although in vitro studies have provided valuable preliminary data, clinical trials are essential for assessing the efficacy of desensitizing treatments, as they replicate the complex biological environment of the oral cavity [13]. Among the differ-ent study designs, the parallel format is more indicated for the evaluation of bioactive materials, as it minimizes the risk of carry-over effects [36]. Furthermore, the type of randomization in a clinical trial is an important factor to be analyzed. In the present study, the randomization performed by conglomerates permitted a homogeneous distribution per group in base-line [25] and after 6 months.

Characteristics of patients and teeth, harmful habits, previous oral treatments, and hygiene instructions may influence the intensity of DH. Most of this data did not directly influence the intensity of initial DH in this study; although, they may have contributed to the development of DH [37]. In this context the first null hypothesis was rejected because the depth of root exposure and presence of antagonist tooth influenced the initial intensity of DH. Smaller root exposure depths were associated with lower levels of dentin sensitivity, as 60.6% of exposures measuring between 0 and 0.4 mm exhibited VAS scores of 6 or lower. A positive correlation between the depth of NCCLs and the intensity of DH was also observed in another clinical trial [38]. In contrast, West et al [29]. reported that recently exposed dentin lesions, even with minimal tooth wear at the cement-enamel junction, may cause intense sensitivity, particularly in young adult individuals. Most teeth had presence of antagonist (96.4%), presenting higher DH levels than teeth without antagonist. 71.4% of teeth without antagonist presented ≤ 5 in VAS. The literature has demonstrated that the development of DH may be associated with occlusal factors; however, recent clinical studies on DH have not evaluated the potential influence of antagonist teeth, when different desensitizers were studied [22,37,39].

The second null hypothesis was also rejected, because all in-office treatments for DH showed a significant reduction in the intensity of sensibility over time compared to baseline. FLU showed a significant reduction in sensitivity at 15 days and its efficacy was sustained from 30 days until 6 months. Clinical studies have shown that this material is capable of reducing DH in the first weeks of follow-up, by the precipitation of calcium fluoride into the dentinal tubules [33,40]. A 6-month randomized clinical trial have reported early symptom reduction that remains relatively constant over follow-up, without substantial improvement between intermediate and late assessments [41].

In an *in vitro* study that evaluated the obliteration of dentinal tubules, this material remained inside the tubules after simulating erosive/abrasive conditions equivalent to a period of 12 months, presenting better results than the placebo [14]. Although FLU has shown effectiveness in reducing hypersensitivity over short-term periods until 6 months [32,42], its therapeutic effect is typically temporary and often requires repeated applications to maintain clinical benefits [3,8].

BIOS showed promising results, with significantly lower sensitivity values at 6 months when compared to baseline, 7 and 15 days in the VAS analysis. Results of *in vitro* studies also suggest that BIOS is able to promote tubular obliteration through hydroxycarbonateapatite deposition, reducing dentin permeability, even after erosive/abrasive challenges [12,14]. A clinical study that evaluated the efficacy and durability of BIOS with different vehicles of application, concluded that this material is able to reduce the intensity of sensibility caused by DH in a short period of time, maintaining this effect at the 6-month follow-up [13]. A scooping review also found that Biosilicate gel and powder achieved significant pain reduction, with minimal recurrence of hypersensitivity at six months [43]. A systematic review concluded that the bioactive glass-based agents demonstrate pronounced benefits in the immediate- to medium-term intervals spanning up to four weeks, however, in the present clinical trial showed a better performance at 6 months follow-up [2].

A reduction of intensity of sensibility was found for SBU after 7 days. This rapid reduction in DH presented by SBU corroborates with other clinical trial, which demonstrated that this material is capable of reducing severe sensitivity, following periodontal surgeries, after the first day of application, maintaining low values for up to 90 days [17]. This fact can be explained because the SBU acts forming a hybrid layer on the dentin tissue, sealing the dentin tubules and promoting a rapid decrease in DH [14,17]. In the present study, patients were able to maintain significantly lower sensitivity values even after 6 months of follow-up, SBU also resist to erosive/abrasive challenge, demonstrated by dentin permeability analysis and confocal microscopy images [14], although there is a lack of clinical studies in the literature that evaluated long-lasting with adhesive system to treat DH. The 10-MDP monomer (10-methacryloyloxydecyl dihydrogen phosphate) present in the material's composition can be responsible for this longevity, since it is capable of binding to the hydroxyapatite, forming calcium salts, improving the mechanical and structural stability of collagen, leading to the dentin matrix stability and consequently reducing the hydraulic conductance [44].

Regarding SPRG, a significant reduction in intensity of sensibility occurred after 15 days. A study that analyzed the same material found significantly reducing DH in a period of 7 and 30 days, for only 20 patients with 60 teeth [18]. SPRG presented the most prolonged effects on DH reduction compared to FLU for up to 8 weeks [20]. *In vitro* studies suggested that this material is able to reduce the permeability of dentin under erosive conditions by mineral deposits on dentin [14,19]. These characteristics are due to the multiple ions present in the composition of the bioactive molecule, being associated with high fluorine release, forming fluorapatite, fluoridated apatite, and/or strontium apatite incorporated into the calcium site in hydroxyapatite [45,46]. These components may promote a buffer capacity, which contributes to the inhibition of the dentin demineralization [45,46]. These factors may have contributed to the longevity of this treatment, which showed significantly lower DH values at 6 months.

When the desensitizers were compared, no significant differences were observed between them in terms of intensity of sensibility, after 6-month follow-up for VAS analysis. However, a difference between FLU and SBU was noted in CoVAS time course after 6-month follow-up. Therefore, the third null hypothesis was also rejected. It should be highlighted that the similar sensibility results among the treatments might be also partially attributed to the positive changes in patient behavior during the study. These effects are likely to happen for all treatments and would interfere in their comparisons.

Some clinical trials also showed similar results for SBU, BIOS, and SPRG desensitizers when compared to the other categories in clinical trials after 1, 3, and 6 months, respectively [13,17,18]. In contrast, other studies found significant differences when compared to other desensitizers, such as Gluma, nano-hydroxyapatite pastes and lasers irradiation [16,24,33,40,42]. SBU demonstrated superior results than other desensitizers in a follow-up of 30 days [47]; however, showed inferior results when compared to other neural action agent [16]. Furthermore, FLU has shown lower results than other desensitizing components in clinical studies of up to 6 months [24,33,40,42].

Regarding the analysis of CoVAS time course, the difference observed between FLU and SBU, where the reaction time for FLU was shorter than for SBU, indicates that the patient has a faster acute response after 15 days and 6 months of using the fluoride-based material. For FLU group, patients exhibited a more acute pain response between 5 and 15 seconds; after that, the response curve continued to rise gradual and stable, reaching its maximum intensity at 30 seconds (Fig 2). In contrast, the SBU group showed a higher increase in sensitivity after 20 seconds of stimulation (Fig 2). These findings corroborated with previous studies that reported better performance of SBU and lower efficacy of FLU in reducing dentin hypersensitivity, particularly when using shorter stimuli to assess acute sensitivity responses [24,33,40,42,47]. A recent *in vivo* study found that the same fluoride varnish used in the present clinical trial provided the least reduction in pain and tubule occlusion, with effects that appeared transient [20]. The literature indicates that adhesive systems are immediately more effective, acting faster in the treatment of DH [17,48]. However, the present study found longer reaction time in CoVAS course after 6 months compared to baseline and 30 days, for the universal adhesive studied. It was noted that the desensitizers showed improvement in reaction time course over time when compared to baseline, with the exception of SPRG. It is important to emphasize that studies with CoVAS reaction time analysis remains limited, which restricted the discussion of the present results with other studies

As a limitation of the present study, it is worth noting that although patients were instructed regarding oral hygiene and the non-use of toothpaste and mouthwash with desensitizing composition, this is a factor that cannot be controlled in a clinical study. Furthermore, only materials that act primarily through the obliteration of dentinal tubules were evaluated. Recent advances in this field have introduced innovative sol-gel technologies to obtain advanced materials to treat DH, such as bioceramics applied in regenerative dentistry [49]. Other materials that rely on different action mechanisms and products should also be compared to the desensitizers evaluated in the present study, for example: depolarization of nerve fibers (potassium-based agents and photobiomodulation therapy) and occlusion of dentin tubules thought albumin precipitation [23,50–55]. In this context, future clinical trials should incorporate these emerging therapies to expand the understanding of available in-office treatment strategies for DH. Additionally, longer-term evaluation and validation in a broader population should be investigated [21].

## Conclusion

Considering the experimental design and findings, this randomized clinical trial demonstrated that height of root exposure and presence of antagonist tooth may influence the initial intensity of DH. All in-office treatments demonstrated sustained reduction in DH over 6 months. Only bioactive resin varnish maintained the same reaction time in all periods. A slowing of the pain reaction time occurred for universal adhesive system when compared to only conventional fluoride varnish.

## Supporting information

**S1 File. Ethics committee approval (Portuguese).** Original ethics committee approval document in Portuguese.
(PDF)

**S2 File. Final report submitted to the ethics committee (Portuguese).** Original final report submitted to the ethics committee in Portuguese.
(PDF)

**S3 File. Study protocol (Portuguese).** Original study protocol in Portuguese, submitted to the ethics committee.
(PDF)

**S4 File. Ethics committee approval (English).** Translated ethics committee approval document in Portuguese.
(PDF)

**S5 File. Final report submitted to the ethics committee (English).** Translated final report submitted to the ethics committee in Portuguese.
(PDF)

**S6 File. Study protocol (English).** Translated study protocol in Portuguese, submitted to the ethics committee.
(PDF)

**S7 File. Clinical trial registration.** Clinical trial registration record in the Brazilian Registry of Clinical Trials (REBEC).
(PDF)

**S8 File. CONSORT 2025 checklist.** Completed CONSORT 2025 checklist for reporting randomized clinical trials.
(PDF)

## Acknowledgments

We wish to thank all the patients that participated in the study.

## Author contributions

**Conceptualization:** André Luiz Fraga Briso, Ticiane Cestari Fagundes.

**Data curation:** Fernanda de Souza e Silva Ramos, Érika Mayumi Omoto, Ticiane Cestari Fagundes.

**Formal analysis:** Fernanda de Souza e Silva Ramos, Érika Mayumi Omoto, Edgar Dutra Zanotto.

**Funding acquisition:** Ticiane Cestari Fagundes.

**Investigation:** Fernanda de Souza e Silva Ramos, Érika Mayumi Omoto, Bruna Perazza, Ticiane Cestari Fagundes.

**Methodology:** Fernanda de Souza e Silva Ramos, Érika Mayumi Omoto, Edgar Dutra Zanotto, Bruna Perazza, Ticiane Cestari Fagundes.

**Project administration:** Ticiane Cestari Fagundes.

**Resources:** Ticiane Cestari Fagundes.

**Supervision:** Ticiane Cestari Fagundes.

**Validation:** Ticiane Cestari Fagundes.

**Visualization:** André Luiz Fraga Briso, Paulo Henrique dos Santos, Ticiane Cestari Fagundes.

**Writing – original draft:** Fernanda de Souza e Silva Ramos, Érika Mayumi Omoto, Bruna Perazza.

**Writing – review & editing:** Fernanda de Souza e Silva Ramos, André Luiz Fraga Briso, Érika Mayumi Omoto, Edgar Dutra Zanotto, Paulo Henrique dos Santos, Bruna Perazza, Ticiane Cestari Fagundes.

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
