## [Decision Letter · Decision Letter 0]

9 Dec 2025

Dear Dr. Fagundes,

Thank you for submitting your manuscript to PLOS ONE. After careful consideration, we feel that it has merit but does not fully meet PLOS ONE’s publication criteria as it currently stands. Therefore, we invite you to submit a revised version of the manuscript that addresses the points raised during the review process.

The topic is clinically relevant, and the study has potential to contribute meaningful evidence regarding the management of dentin hypersensitivity. We appreciate your efforts in designing and conducting this clinical trial. However several **major methodological, statistical, and reporting issues** that must be addressed before the manuscript can be considered for publication.

The main issue is discrepancy between:

the CONSORT flowchart (subject-level allocation), andthe methods section (stratified randomization at the tooth level)

Please clarify whether randomization occurred at tooth level or patient level, adjust CONSORT diagram and text to ensure full consistency and explain allocation procedures accordingly.

Since baseline VAS score and exposure height were used as stratification variables, they must also be incorporated into the statistical models. Please review the statistical method of this clinical trial.

We look forward to receiving your revised manuscript.

Kind regards,

Esra Cengiz Yanardag

Academic Editor

PLOS One

Journal Requirements:

“We wish to thank all the patients that participated in the study. This study was financed in part by the Coordenação de Aperfeiçoamento de Pessoal de Nível Superior – Brasil (CAPES) -Finance Code 001. This research was also supported by the São Paulo Research Foundation—FAPESP [research grant number 2020/07625-0 and scholarship grant number 2020/07443-9]”

“This study was financed in part by the Coordenação de Aperfeiçoamento de Pessoal de Nível Superior – Brasil (CAPES) -Finance Code 001. This research was also supported by the São Paulo Research Foundation—FAPESP [research grant number 2020/07625-0 and scholarship grant number 2020/07443-9]”

“This study was financed in part by the Coordenação de Aperfeiçoamento de Pessoal de Nível Superior – Brasil (CAPES) -Finance Code 001. This research was also supported by the São Paulo Research Foundation—FAPESP [research grant number 2020/07625-0 and scholarship grant number 2020/07443-9]”

4. We note that the original protocol that you have uploaded as a Supporting Information file contains an institutional logo. As this logo is likely copyrighted, we ask that you please remove it from this file and upload an updated version upon resubmission.

5. Please include captions for your Supporting Information files at the end of your manuscript, and update any in-text citations to match accordingly. Please see our Supporting Information guidelines for more information: http://journals.plos.org/plosone/s/supporting-information .

Reviewers' comments:

Reviewer's Responses to Questions

**Comments to the Author**

1. Is the manuscript technically sound, and do the data support the conclusions?

Reviewer #1: No

Reviewer #2: Yes

Reviewer #3: Yes

2. Has the statistical analysis been performed appropriately and rigorously?

Reviewer #1: No

Reviewer #2: Yes

Reviewer #3: Yes

3. Have the authors made all data underlying the findings in their manuscript fully available?

Reviewer #1: Yes

Reviewer #2: Yes

Reviewer #3: Yes

4. Is the manuscript presented in an intelligible fashion and written in standard English?

Reviewer #1: Yes

Reviewer #2: Yes

Reviewer #3: Yes

Reviewer #1: This manuscript reports findings from a single-blinded randomized trial involving four parallel desensitizing treatment groups. While the topic is relevant and the study design has potential, the current version requires major revisions and clarifications before it can be considered for publication.

Major Revisions

1. The study compares four different desensitizing treatments without including an active control group. Therefore, the conclusion that all four in-office treatments are effective is not supported by the design.

2. The CONSORT flowchart suggests that each subject was assigned to a treatment group, whereas lines 134–135 indicate stratified randomization at the tooth level. This discrepancy needs clarification.

3. If stratified randomization was applied, baseline VAS score and exposure height (used as stratification factors) should be incorporated into all statistical analyses.

4. The sample size calculation lacks essential details, including assumptions about variance and the specific statistical test employed. The text implies an expected mean VAS difference of 0.21, but the corresponding variance is not provided. Additionally, with four treatments under study, adjustments for multiple comparisons should be addressed.

5. Line 135 states that the tooth was the unit of analysis. If multiple teeth per subject were analyzed, intra-subject correlation must be accounted for. Standard linear regression is inappropriate in this context.

6.Line 183 suggests that linear mixed-effects models were used. The manuscript should specify:

Random effects included

Correlation/dependence structure

Assessment of normality assumptions

Any data transformations applied to meet model assumptions

7. CoVAS data are longitudinal. It is unclear how Kruskal-Wallis was applied—was it performed at each time point using cross-sectional data? If so, adjustments for multiple testing should be considered.

Minor Revisions:

1. Tables 2 and 3 use commas for decimal points; these should be replaced with periods.

2. Superscripts (a–d) in Table 3 require clearer explanation. Current footnotes are confusing.

3.Root exposure and tooth type appear to be tooth-specific, yet Table 2 presents subject-level data. This inconsistency needs resolution.

4.The phrase “no intention to be treated was used” is unclear and should be revised for clarity.

Reviewer #2: This longitudinal, randomized, parallel-design clinical trial aimed to evaluate the long-term effectiveness of different in-office treatments for dentin hypersensitivity over a 6-month period. Although it presents an interesting idea, some points need to be clarified:

INTRODUCTION

Solid scientific foundation, with several current and relevant references.

The introduction follows a logical progression, from the clinical definition of the problem to the justification for the study. It clearly explains the physiological mechanisms and therapies involved.

Although the study evaluated the influence of characteristics related to root exposure on the initial intensity of dentin hypersensitivity, this analysis was not mentioned among the stated objectives. It is recommended that this variable be explicitly included in the objectives to align the methodological design with the presentation of the results and conclusions of the study.

MATERIALS AND METHODS

There is a need to clarify the blinding of the study

The treatments involved materials with distinct application procedures and physical characteristics (e.g., viscosity, application time, and photopolymerization requirements), therefore, patient blinding was not feasible.

Did the researcher who statistically analyzed the results know the groups evaluated?

There is a need to clarify the time period used in the CoVas analysis.

Page8, line 158- Interventions: Was there standardization regarding the brushing technique and the toothpaste used daily? It is important to clarify.

RESULTS

Page 12, line 223- I suggest differentiating the intergroup analyses with capital letters and the intragroup analyses with lowercase letters, or vice versa, as it makes it easier to understand the analysis of the results. This way, it became confusing to interpret the data.

Page12, line 224- Describe which test was used.

DISCUSSION

Page 16, paragraph (line 332-343) - The authors did not specify, in the methodology section, the periods of evaluation of dentin hypersensitivity, which makes it difficult to understand and interpret the results discussed.

CONCLUSION

I suggest adjusting the conclusion to the objectives.

UPDATED REFERENCES

Reviewer #3: This manuscript reports a 6-month clinical trial evaluating the efficacy of a desensitizing agent for managing dentin hypersensitivity. The topic is clinically significant and relevant. The manuscript is well prepared and suitable for publication. I just have a few minor comments:

-Please clarify whether participants used any desensitizing toothpaste or mouthrinse before or during the study. This could influence the sensitivity outcomes and should be mentioned as a possible limitation.

-Review abbreviation footer in Table 1. Some abbreviations used in the table are not listed while others listed there do not appear in the table.

-The need for longer-term evaluation and validation in a broader population should be highlighted.

Thank you!

**Do you want your identity to be public for this peer review?** For information about this choice, including consent withdrawal, please see our Privacy Policy

Reviewer #1: No

Reviewer #2: No

Reviewer #3: No

---

## [Author Response · Author response to Decision Letter 1]

12 Jan 2026

We would like to thank the reviewers for the careful and constructive review of our manuscript. Below, we provide the responses. The reviewers’ comments are in bold and our answers are presented immediately below. We have submitted a revised version of the manuscript with changes highlighted in yellow (Reviewer 1), green (Reviewer 2), and orange (Reviewer 3). All changes made because changes in results due to new statistical analyses are highlighted in pink

Reviewer #1

Question R1.1- This manuscript reports findings from a single-blinded randomized trial involving four parallel desensitizing treatment groups. While the topic is relevant and the study design has potential, the current version requires major revisions and clarifications before it can be considered for publication. Major Revisions

R1.1: We thank for your constructive comments and careful evaluation of our manuscript. As requested, we have thoroughly revised the manuscript, providing the clarifications. The main revisions are detailed point-by-point in the responses below and in the manuscript highlighted in yellow.

Questions R1.2- The study compares four different desensitizing treatments without including an active control group. Therefore, the conclusion that all four in-office treatments are effective is not supported by the design.

R1.2: We thank the reviewer for this comment. The fluoride varnish group (FLU) was an active control, as they are widely studied and commonly used as in-office management of dentin hypersensitivity. The inclusion of an untreated control group was not permitted due to ethical considerations, since all participants presented dentin hypersensitivity associated with pain. Therefore, our conclusions were based on comparisons with an established active control; however, to improve clarity, the FLU group was explicitly identified as the active control in summary and subitem “Sample size, subjects’ selection and groups” as follows: “...A total of 68 patients were screened; 42 patients (n=192 teeth) were included. Participants were randomly assigned to one of four treatment groups: fluoride varnish (FLU, active control), biosilicate solution (BIOS), universal adhesive system (SBU), or varnish containing S-PRG particles (SPRG), (Table 1)...”

Questions R1.3-. The CONSORT flowchart suggests that each subject was assigned to a treatment group, whereas lines 134–135 indicate stratified randomization at the tooth level. This discrepancy needs clarification.

R1.3: Thank you so much for pointing out this discrepancy in our manuscript. We checked this discrepancy with all authors involved in this clinical trial. We forgot to insert about those patients with more than one eligible tooth, classification was based on the mean of exposure heights and the mean of VAS sensitivity scores. Therefore, randomization was conducted at the patient level rather than at the tooth level, as this was a parallel-group study design. This information has been incorporated as follows: “…Patients who had more than one eligible tooth, classification was based on the mean of exposure heights and the mean of VAS sensitivity scores…”

Questions R1.4- If stratified randomization was applied, baseline VAS score and exposure height (used as stratification factors) should be incorporated into all statistical analyses.

R1.4: This is also an important comment. The statistical analysis was done again, including this recommendation. Although there was a p value of 0.01 (after 6 months) in the distribution of strata between the groups does not compromise the validity of the randomization, as it is compatible with random fluctuations expected in small samples. The inclusion of the stratification variable as a covariate in the mixed-effects models allowed adequate control of this factor, reducing the risk of bias and ensuring robust statistical inferences. It is important to emphasize that the p value was 0.07 at baseline, regarding the distribution of strata between the groups.

Questions R1.5-. The sample size calculation lacks essential details, including assumptions about variance and the specific statistical test employed. The text implies an expected mean VAS difference of 0.21, but the corresponding variance is not provided. Additionally, with four treatments under study, adjustments for multiple comparisons should be addressed.

R1.5: Thank you for this comment. The Sigma Plot 12.0 software (Systat Software Inc., London, UK) was used to calculate the sample size. This software describes ANOVA sample size as the statistical test employed. The statistician redone the analyses and she found an error because she used a desired power of 0.73. We checked that this power is reasonable (Sihoe AD. Rationales for an accurate sample size evaluation. J Thorac Dis. 2015 Nov;7(11):E531-6. doi: 10.3978/j.issn.2072-1439.2015.10.33.) Additionally, we included a print screen from this software to observe what it asks during the process. The text was modified to describe the content of this calculation: “…Sample size calculation was based on data from a previous study, [24] using analysis of variance (ANOVA) sample size (Sigma Plot 12.0, Systat Software Inc., London, UK). The analysis assumed a minimum detectable difference in means of 0.210 (expected standard deviation of residuals of 0.3) on the Visual Analogue Scale (VAS), with a significance level (α) of 0.05, desired power of 0.73, and estimated dropout rate of 20.0%. The calculation resulted in initial sample size of 48 teeth per group. ….”

Questions R1.6- Line 135 states that the tooth was the unit of analysis. If multiple teeth per subject were analyzed, intra-subject correlation must be accounted for. Standard linear regression is inappropriate in this context.

Questions R1.7- Line 183 suggests that linear mixed-effects models were used. The manuscript should specify: Random effects included Correlation/dependence structure. Assessment of normality assumptions Any data transformations applied to meet model assumptions.

Questions R1.8- CoVAS data are longitudinal. It is unclear how Kruskal-Wallis was applied—was it performed at each time point using cross-sectional data? If so, adjustments for multiple testing should be considered.

R1.6, 7, 8: We thank to point out these inconsistencies in the statistical analysis. We asked the statistician to revise all the statistical analysis based on your corrections. The statistician sent us the new methodology used to perform the analysis. Please find below the methods used now:

- In tables 2, 3 and 4 comparisons involving qualitative variables among groups were performed using Fisher’s exact test, as well as comparisons involving quantitative variables among groups were performed using analysis of variance (ANOVA).

- In table 5 and figure 2 comparisons of quantitative outcomes (VAS and CoVAS time course), linear mixed-effects regression models including fixed and random effects were employed. Linear mixed-effects models are appropriate for the analysis of clustered data, in which more than one observation may originate from the same individual and the assumption of independence between observations is violated (Schall, 1991). Random effects consisted of subject-specific random intercepts, allowing the correlation between repeated measurements from the same participant to be appropriately modeled. A variance components covariance structure was adopted, assuming different variances for the random effects and no correlation among them.

- In this context, all models were adjusted for the initial stratification variables used during randomization, ensuring adequate control of these factors in the study design (the distribution of initial stratification variables used during randomization was inserted in the table 2). Model assumptions, particularly the normality of residuals, were evaluated through graphical inspection using histograms, quantile–quantile (Q–Q) plots, and residual-versus-fitted plots.

- Group and time comparisons were performed using post hoc orthogonal contrasts. Given the longitudinal design with repeated measures and the prespecified and independent nature of the orthogonal contrasts, no additional correction for multiple comparisons was applied, as this approach minimizes the risk of inflated type I error.

- Statistical analyses were conducted using SAS software (version 9.4). A significance level of 5% was adopted for all analyses.

- The Statistical Analysis section was revised, summarizing the above text: “... Comparisons involving qualitative variables among groups were performed using Fisher’s exact test, as well as comparisons involving quantitative variables were performed using ANOVA (Tables 2, 3 and 4). Comparisons of quantitative longitudinal outcomes (VAS and CoVAS time course), linear mixed-effects regression models including fixed and random effects were employed (Table 5 and Fig 2). A variance components covariance structure was adopted. All models were adjusted for the initial stratification variables used during randomization, ensuring adequate control of these factors in the study design. Model assumptions, particularly the normality of residuals, were evaluated through graphical inspection using histograms, quantile–quantile (Q–Q) plots, and residual-versus-fitted plots. Group and time comparisons were performed using post hoc orthogonal contrasts. No additional correction for multiple comparisons was applied. Statistical analyses were conducted using SAS 9.4 software (SAS Institute Inc. NC, USA).

A per-protocol analysis was conducted, including only participants who completed all evaluation timepoints. All data were considered statistically different if p≤0.05)...”

Questions R1.9- Minor Revisions: Tables 2 and 3 use commas for decimal points; these should be replaced with periods.

R1.9: Thank you so much. The decimal points were inserted in Tables 2 and 3 (now tables 2, 3 and 4 to address comment R1.11).

Questions R1.10- Superscripts (a–d) in Table 3 require clearer explanation. Current footnotes are confusing.

R1.10: Thank you for this comment; however, Table 3 does not have any superscript letters. Then, we think that you referred to Table 4 (now Table 5 to address comment R1.11). Uppercase letters were included to indicate no differences among materials within the same analysis time, as follow: “…Uppercase letters indicate significant differences among materials, within the same analysis times. Lowercase letters indicate significant differences among analysis times, within the same material…”

Questions R1.11- Root exposure and tooth type appear to be tooth-specific, yet Table 2 presents subject-level data. This inconsistency needs resolution.

R1.11: We appreciated the reviewer’s suggestion. The descriptive data has been reorganized into two separate tables. Table 2 presents patient-level characteristics and Table 3 presents teeth-level characteristics.

Questions R1.12-.The phrase “no intention to be treated was used” is unclear and should be revised for clarity.

R1.12: The term was replaced with the appropriate terminology for this type of analysis. The revised sentence is: “…A per-protocol analysis was conducted, including only participants who completed all evaluation timepoints....”

Reviewer #2:

Questions R2.1- This longitudinal, randomized, parallel-design clinical trial aimed to evaluate the long-term effectiveness of different in-office treatments for dentin hypersensitivity over a 6-month period. Although it presents an interesting idea, some points need to be clarified:

R2.1: We for highlighting important points that require clarification. In accordance with the reviewer’s suggestions, we have revised the manuscript to address each issue in detail. All suggestions are highlighted in green in the manuscript.

Questions R2.2- INTRODUCTION Solid scientific foundation, with several current and relevant references. The introduction follows a logical progression, from the clinical definition of the problem to the justification for the study. It clearly explains the physiological mechanisms and therapies involved. Although the study evaluated the influence of characteristics related to root exposure on the initial intensity of dentin hypersensitivity, this analysis was not mentioned among the stated objectives. It is recommended that this variable be explicitly included in the objectives to align the methodological design with the presentation of the results and conclusions of the study.

R2.2: Thank you for the positive evaluation of the Introduction section and for the valuable suggestion. We agree that the influence of patient and teeth characteristics should be explicitly stated in the study objectives. Accordingly, the following sentence was added to the objectives: “…Additionally, the influence of characteristics related to root exposure on the initial intensity of dentin hypersensitivity was also evaluated. …”.

Questions R2.3- MATERIALS AND METHODS - There is a need to clarify the blinding of the study. The treatments involved materials with distinct application procedures and physical characteristics (e.g., viscosity, application time, and photopolymerization requirements), therefore, patient blinding was not feasible. Did the researcher who statistically analyzed the results know the groups evaluated?

R2.3: We thank the reviewer for this important question. Although the materials differed in viscosity, application time, and the need for photopolymerization, patient blinding was maintained. Each patient received treatment individually and was not informed about the study groups or the characteristics of the materials, preventing them from identifying which intervention was applied. Regarding the blinding of the outcome assessment, the statistician who performed the data analysis was fully blinded to the treatment allocation. The groups were coded and provided to the statistician using Roman numerals (I, II, III, and IV), corresponding to FLU, BIOS, SBU, and SPRG, respectively, without any indication of the materials used.

Questions R2.4- There is a need to clarify the time period used in the CoVas analysis.

R2.4: We appreciate the reviewer’s comment. The text has been revised, including examples, as follow to improve clarity: “…During stimulation, the participant continuously adjusted a manually controlled potentiometer to indicate the perceived sensibility on a 0–100 scale. [30] For each assessment, the maximum sensibility score and the exact time (in seconds) were recorded. Additionally, if the participant reached a score of 100, the stimulus was immediately discontinued and the corresponding time was recorded. For example, if a participant reported a maximum score of 13 at 27 seconds and this value remained constant until 30 seconds, the recorded values were 13 and 27 seconds. However, if a participant reached a pain score of 100 at 1 second the stimulus was stopped immediately, and these two values were registered…”

Questions R2.5- Page 8, line 158- Interventions: Was there standardization regarding the brushing technique and the toothpaste used daily? It is important to clarify.

R2.5: Thank you for your observation. Patients received standardized oral hygiene instructions at baseline. They were advised to use a conventional toothpaste without desensitizing agents throughout the study period. This information was added to the revised manuscript: “…Patients were instructed regarding oral hygiene instructions, being advised the non-use of toothpaste and mouthwash with desensitizing composition…

Questions R2.6- RESULTS- Page 12, line 223- I suggest differentiating the intergroup analyses with capital letters and the intragroup analyses with lowercase letters, or vice versa, as it makes it easier to understand the analysis of the results. This way, it became confusing to interpret the data.

R2.6: Thank you for the suggestion. Capital letters were added to represent intergroup comparisons.

Questions R2.7- Page12, line 224- Describe which test was used.

R2.7: We thank the reviewer for the suggestion. This sentence was inserted in the footnote: “…Linear mixed-effects regression models including fixed and random effects were employed (p≤0.05).”

Questions R2.8- DISCUSSION - Page 16, paragraph

---

## [Decision Letter · Decision Letter 1]

27 Jan 2026

Longevity of different in-office treatments for dentin hypersensitivity: a 6-month randomized and parallel clinical trial

PONE-D-25-49127R1

Dear Dr. Fagundes

We’re pleased to inform you that your manuscript has been judged scientifically suitable for publication and will be formally accepted for publication once it meets all outstanding technical requirements.

Kind regards,

Esra Cengiz Yanardag

Academic Editor

PLOS One

Additional Editor Comments (optional):

The revisions made in response to the reviewers’ comments, particularly those concerning the statistical analyses, have been carefully evaluated. The authors have adequately addressed all major and minor points raised during the review process, and that the manuscript has been substantially improved as a result.

Reviewers' comments:

Reviewer's Responses to Questions

**Comments to the Author**

Reviewer #1: All comments have been addressed

Reviewer #2: All comments have been addressed

2. Is the manuscript technically sound, and do the data support the conclusions?

Reviewer #1: Yes

Reviewer #2: Yes

3. Has the statistical analysis been performed appropriately and rigorously?

Reviewer #1: Yes

Reviewer #2: Yes

4. Have the authors made all data underlying the findings in their manuscript fully available?

Reviewer #1: Yes

Reviewer #2: Yes

5. Is the manuscript presented in an intelligible fashion and written in standard English?

Reviewer #1: Yes

Reviewer #2: Yes

Reviewer #1: All comments have been carefully and fully addressed by the author team and hence it is recommended to accept for publicaction.

Reviewer #2: The authors adequately addressed all comments and suggestions presented in the previous review. Their responses were clear, well-founded, and supported by appropriate revisions throughout the manuscript. The implemented changes improved methodological clarity, enhanced the presentation of results, and strengthened the discussion, thus aligning the manuscript with expected scientific standards. I consider the manuscript adequately revised and suitable to proceed in the editorial process. The revisions made by the authors significantly improved the methodological quality of the study, enhanced the clarity of the presentation of results, and strengthened the discussion. Therefore, the manuscript meets the required scientific standards and can be considered suitable for publication in PLOS ONE.

**Do you want your identity to be public for this peer review?** For information about this choice, including consent withdrawal, please see our Privacy Policy

Reviewer #1: No

Reviewer #2: No

---

## [Editor Report · Acceptance letter]

PONE-D-25-49127R1

PLOS One

Dear Dr. Fagundes,

I'm pleased to inform you that your manuscript has been deemed suitable for publication in PLOS One. Congratulations! Your manuscript is now being handed over to our production team.

Kind regards,

on behalf of

Dr. Esra Cengiz Yanardag

Academic Editor

PLOS One